# RNA m6a Methylation Regulator Expression in Castration-Resistant Prostate Cancer Progression and Its Genetic Associations

**DOI:** 10.3390/cancers16071303

**Published:** 2024-03-27

**Authors:** Chamikara Liyanage, Achala Fernando, Audrey Chamberlain, Afshin Moradi, Jyotsna Batra

**Affiliations:** 1School of Biomedical Sciences, Faculty of Health, Queensland University of Technology, Brisbane, QLD 4059, Australiaa.vitharanage@qut.edu.au (A.F.);; 2Center for Genomics and Personalized Health, Translational Research Institute, Queensland University of Technology, Brisbane, QLD 4059, Australia

**Keywords:** prostate cancer, castration-resistant prostate cancer, androgen-targeted therapy, m6A methylation, single-nucleotide polymorphisms

## Abstract

**Simple Summary:**

The study investigates N6-methyladenosine (m6A) methylation, the most common RNA modification that influences cell function by regulating RNA’s function and metabolism. The reversible control of m6A modification involves methyltransferases (“writers”) and demethylases (“erasers”), with the interactions facilitated by RNA-binding proteins (“readers”). The dysregulation of these processes is implicated in various diseases, including cancers. Using mass spectrometry and RNA sequencing, this study profiles the dynamic m6A regulator expression patterns in prostate cancer. It identifies specific regulators whose expression significantly changes during cancer development, therapy resistance, and progression to advanced stages like castration-resistant prostate cancer. The research also uncovers significant correlations between dysregulated m6A regulators and genetic polymorphisms linked to prostate cancer risk. In conclusion, this study underscores the clinical importance of m6A-regulators, offering potential molecular targets for diagnostic and prognostic assessments in prostate cancer.

**Abstract:**

N6-methyladenosine (m6A) methylation, a prevalent epitranscriptomic modification, plays a crucial role in regulating mRNA expression, stability, and translation in mammals. M6A regulators have gained attention for their potential implications in tumorigenesis and clinical applications, such as cancer diagnosis and therapeutics. The existing literature predominantly addresses m6A regulators in the context of primary prostate cancer (PCa). However, a notable gap in the knowledge emerges regarding the dynamic expression patterns of these regulators as PCa progresses towards the castration-resistant stage (CRPC). Employing sequential window acquisition of all theoretical mass spectra (SWATH-MS) and RNAseq analysis, we comprehensively profiled the expression of 27 m6A regulators in hormone/androgen-dependent and -independent PCa cell lines, revealing distinct clustering between tumor and adjacent normal prostate tissues. High-grade PCa tumors demonstrated the upregulation of *METTL3*, *RBM15B*, and *HNRNAPA2B1* and the downregulation of *ZC3H13*, *NUDT21*, and *FTO*. Notably, we identified six m6A regulators associated with PCa survival. Additionally, association analysis of the PCa-associated risk loci in the cancer genome atlas program (TCGA) data unveiled genetic variations near the *WTAP*, *HNRNPA2B1*, and *FTO* genes as significant expression quantitative trait loci. In summary, our study unraveled abnormalities in m6A regulator expression in PCa progression, elucidating their association with PCa risk loci. Considering the heterogeneity within the PCa phenotypes and treatment responses, our findings suggest that prognostic stratification based on m6A regulator expression could enhance PCa diagnosis and prognosis.

## 1. Introduction

Prostate cancer (PCa) is the second most frequent cancer and the fifth leading cause of cancer death among men [1]. The estimated rates for PCa diagnosis and mortality are set to almost double by 2040, simply due to the growth and aging of the global population [2]. Due to PCa’s dependence on oncogenic androgen receptor (AR) signaling, androgen deprivation therapy (ADT) is considered the primary systemic therapy for patients with localized, recurrent, or advanced-stage disease. ADT is often supplemented with androgen-targeted therapy (ATT), with alternative methods to suppress the activation of androgen receptor (AR) signaling [3]. Yet nearly all patients treated with ADT/ATT, with or without additional systemic therapy, will progress to an aggressive stage, termed hormone-independent/castration-resistant PCa (CRPC) [4]. Over 85% of CRPC patients develop metastatic CRPC (mCRPC), indicating nodal, bone, and/or visceral spread and showing a median survival range from ~18 to ~36 months [4,5]. Thus, the advancement of omics-based technologies and a better understanding of tumor biology and multidrug resistance are critical for the development of novel diagnostic and therapeutic agents for PCa.

The chemical modification of biological macromolecules is a highly specific regulatory mechanism in many diseases, including cancer [6,7,8,9]. Currently, more than 170 RNA modifications have been identified, composing the human “epitranscriptome” [10]. These RNA modifications can regulate the RNA’s post-transcriptional stability, localization, translocation, splicing, and translation and participate in multiple biological processes and disease onsets. The methylation of adenosine at position 6, resulting in N6-methyladenosine (m6A), is the most characterized and abundant RNA modification in all eukaryotes [11,12]. Studies have shown that m6A methylation exists on more than 7000 coding and 300 non-coding RNAs, where 0.1–0.4% of the total adenine transcriptomic nucleotides are methylated in mammals [13,14]. Like DNA methylation, m6A methylation is reversibly regulated by the following enzymes: m6A methyltransferases (METTL3/14, WTAP, RBM15/15B, VIRMA, and ZC3H13, termed “writers”) and m6A demethylases (FTO, ALKBH5, and ALKBH3, termed “erasers”). M6A methylation executes its primary function via interaction with RNA-binding proteins (YTHDC1/2, YTHDF1/2/3, IGF2BP1/2/3, HNRNPs, termed “readers”). These enzymes dynamically regulate m6A-dependent molecular processes and physiological functions and their aberrancies, frequently found in a variety of diseases and malignancies, such as gastric cancer, colorectal cancer, thyroid cancer, breast cancer, pancreatic cancer, kidney cancer, sarcoma, leukemia, PCa, etc. [8,9].

Although the evidence highlights aberrant m6A regulation patterns in multiple cancers, the mechanism underlying the regulatory role of m6A modification in cancer progression remains incompletely understood. The overexpression of m6A regulators has been implicated in tumorigenesis in lung cancer, breast cancer, and certain gynecological tumors, yet conflicting roles of m6A regulators persist in other cancer types [6,7,15]. Therefore, precise profiling of m6A-regulator-based signatures is crucial for investigating the diverse regulatory impact of m6A regulators on cancer progression, survival, and drug resistance. In terms of PCa, the previous literature links PCa pathogenesis and progression to key m6A regulators, such as METTL3 [16], VIRMA [17], YTHDF2/3 [18,19] FTO [20], and RBM15 [21]. Yet, these studies lack evidence on treatment-induced m6A regulator signatures and their association with CRPC progression. Here, we provide comprehensive proteomic and transcriptomic evidence of differential m6A regulation in different PCa cell phenotypes and in response to ATT treatments. We further profile the m6A-regulator-based signature in PCa clinical specimens and its association with GWAS-identified PCa risk loci. Finally, using publicly available data, we revealed the key m6A regulators that may facilitate PCa progression to CRPC and their associated risk with a poor PCa prognosis.

## 2. Materials and Methods

### 2.1. Cell Culture

The androgen-dependent/sensitive LNCaP and DUCaP cell lines and androgen-independent PC3, DU145, and C4-2B cell lines were procured from the American Type Culture Collection and confirmed to be negative for mycoplasma contamination. All the cell lines were cultured in RPMI 1640 media (Life Technologies, Grand Island, NY, USA), supplemented with either 5% or 10% fetal bovine serum (FBS) (Sigma-Aldrich, Life Technologies, Thornton, NSW, Australia), and incubated at 37 °C with 5% CO_2_ until they reached 100% confluency.

For androgen deprivation and androgen/anti-androgen treatments in PCa cell lines, the LNCaP cells were seeded into 6-well plates with media and incubated at 37 °C with 5% CO_2_ for 48 h. The media were then replaced with androgen-depleted media (RPMI 1640) supplemented with 5% charcoal-stripped serum (CSS) and incubated for an additional 48 h at 37 °C with 5% CO_2_. Subsequently, the cells were treated with the following conditions: (i) 10 nM androgens/dihydrotestosterone (DHT) (Sigma-Aldrich, Sydney, NSW, Australia), (ii) 10 µM anti-androgens, namely Bicalutamide (BIC) or Enzalutamide (ENZ) (Selleckchem.com, Waterloo, NSW, Australia), (iii) a combination of both androgens and anti-androgens (DHT + BIC, DHT + ENZ), and (iv) 20% ethanol (EtOH) as the vehicle control (HPLC-grade, Sigma-Aldrich). All the treated cells were further incubated for 48 h at 37 °C with 5% CO_2_.

### 2.2. RNA Isolation and Quantitative PCR Analysis of the Formalin-Fixed, Paraffin-Embedded (FFPE) Tissues and PCa Cell Lines

Thirty-five FFPE prostatic tissues and their corresponding adjacent normal tissues were obtained from PCa patients through the Australian PCa BioResource (APCB). All the patients had signed the consent forms, and the study was approved by Metro South Health and the QUT ethics board (Approval no 1000001165 Approval Date: 30 May 2023). The age range of the PCa patients was between 52 and 65 years, and the Gleason score ranged from 6 to 9.

The tissue blocks containing the tumor cells were serially sectioned (20 μm), transferred to glass slides, and stained with hematoxylin and eosin (H&E). Two pathologists marked the tumor areas, and these marked areas were manually dissected using a sterile injection needle under a microscope. Deparaffinization of the dissected tissues was carried out using deparaffinization solution (QIAGEN, Clayton, VIC, Australia, Catalog number—19093). RNA extraction from the tissue samples was performed using the miRNeasy FFPE kit (QIAGEN). Additionally, RNA isolation from the PCa cell lines was carried out using the RNeasy Mini Kit following the standard protocol (QIAGEN). The purity and concentration of the extracted RNA samples were assessed using NanoDrop™1000 (Thermo Scientific, Waltham, MA, USA). Quantitative PCR analysis was performed using the comparative (ΔΔCT) method.

### 2.3. Sequential Window Acquisition of All THeoretical Mass Spectra (SWATH-MS) Analysis

Cell pellets were obtained from the PCa cell lines (LNCaP, DUCaP, C4-2B, PC3, DU145) and the androgen- and anti-androgen-treated LNCaP cell lines. The protein lysates were obtained using sodium deoxycholate (SDC) lysis buffer (1% SDC in 1M Tris pH 8.0). To remove cell debris, the samples were centrifuged at 16,000 rpm for 20 min. To denature the proteins and shear the DNA, the samples were sonicated in an ultrasonic bath (Thermo Scientific™) for 15 min (at 4 °C, 100% Power) The concentration of the proteins was calculated using a Bicinchoninic acid assay (BCA) with Pierce™ Bovine Serum Albumin (BSA) standard (Thermo Scientific™). A total of 30 μg of protein was denatured at 95 °C for 5 min using a thermomixer (ThermoMixer^®^ F1.5, Eppendorf, Hamburg, Germany). The subsequent samples with denatured proteins were reduced by adding 10 mM Tris (2-carboxyethyl) phosphine (Sigma-Aldrich), kept for 30 min at room temperature, and alkylated by adding 40 mM 2-chloroacetamide (2CAA) (Sigma-Aldrich) for 30 min in the dark at room temperature. Trypsin (Sigma-Aldrich) was added at a 1:50 enzyme–protein ratio to digest the proteins overnight at 37 °C. The digestion reaction was inhibited by adding 10% trifluoroacetic acid (TFA) (Sigma-Aldrich), and Pierce™ C18 Spin Tips (Thermo Fisher) were used for peptide desalting and washed in 0.1% TFA and eluted into 80% acetonitrile (ACN) (Sigma-Aldrich). The peptides were concentrated by evaporating the solvents using a SpeedVac centrifuge (Savant SPD121P-230 SpeedVac, hermo Electron Corporation, Milford, MA, USA) at 35 °C and re-suspending them in iRT calibration mix comprising 2% ACN and 0.1% TFA (Biognosys AG, Schlieren, Switzerland). All the samples were submitted for the LC-MS/MS analysis. In 3 biological replicates, LC-MS/MS data-dependent acquisition (DDA), data-independent acquisition (DIA), and data analysis for the protein expression were performed according to our previously published protocols for SWATH-MS analysis in PCa cell lines [22,23]. The SWATH-MS peak areas for each protein target were statistically analyzed using the MetaboAnalyst 5.0 software. Only m6A regulators identified at <1% FDR and >90% confidence in the PCa cell lines were considered for the analysis. The peak area values for each protein obtained from the three replicates were averaged, normalized (based on the sum), and log_2_-transformed. *p* < 0.05 was considered significant.

### 2.4. RNAseq Analysis for the PCa Cell Lines and PCa Clinical Tissues

The RNAseq analysis performed in our previously published study was reanalyzed to determine the androgen- and anti-androgen-mediated m6A regulator expression in LNCaP cells [24]. According to the study, the total RNA extracted from the androgen/anti-androgen-treated LNCaP cell lines and eight clinical prostate tumors and their adjacent non-malignant tissues was used for the RNAseq analysis, performed through the Australian Genome Research Facility (AGRF), as mentioned in our previously published protocol [25].

### 2.5. RNA Expression Analysis Using Published Transcriptomic Data

The differential expression of the m6A methylation regulators in the localized PCa and CRPC PCa clinical data was revaluated by measuring their transcriptomic expression in the Grasso PCa clinical microarray datasets obtained from the Oncomine™ database (https://www.oncomine.org (accessed on 15 May 2019)).

### 2.6. Survival Analysis

The prognostic significance of the m6A methylation regulators was analyzed using Kaplan–Meier plotter (http://kmplot.com/analysis/index (accessed on 20 April 2021)) [26]. Kaplan–Meier plotter analyzed the correlation between the expression of each m6A regulator and the overall survival of the CRPC patients after the 1st hormone therapy. The parameters used were as follows: *p*-value < 0.05; split patients by: Auto select best cutoff; Follow up threshold: All; Cox regression: univariate.

### 2.7. Cis-Expression Quantitative Trait Loci (eQTL) Analysis

We obtained the genotype and RNA data for 496 prostate cancer patients from the TCGA data portal (https://tcga-data.nci.nih.gov/tcga/ (accessed on 5 August 2019)). For RNAseq, we employed the HISAT2 program to align the sequencing reads with the human transcriptome, and quantification was performed using feature-count [27]. Quality control for the SNPs was conducted using the genotype data, where individuals with more than 2% missing genotypes were excluded. SNPs with a call rate < 98% and those not in Hardy–Weinberg equilibrium (*p* < 10^−8^) were excluded. Additionally, SNPs with a minor allele frequency < 5% were excluded from further analysis. Samples with a high heterozygosity (mean heterozygosity ± 3 SD) were filtered out. Imputation of the samples was carried out using the 1000 Genomes Project phase 3 reference panel. Population stratification was addressed, and individuals identified as outliers in the European population (6 SD) were removed. Ultimately, 386 PCa patients of European ancestry possessing both genotype and RNAseq data were selected for the eQTL analysis. The Cis-eQTL analysis between the gene expression and cleaned genotypes was performed using ANOVA, selecting the SNP with the lowest *p*-value for a particular gene.

## 3. Results

### 3.1. Differential Expression of m6A Methylation Regulators in PCa Cell Lines

The protein expression of 7 m6A writers, METTL3/14, WTAP, VIRMA, RBM15/15B, ZC3H13; 18 m6A readers, IGF2BP2/3, YTHDF1/2/3, YTHDC1/2, HNRNPC, HNRNPA2B1, RBMX, PRRC2A, CPSF6, NXF1, NUDT21, SRSF10, TRMT112, XRN1, FMR1; and 2 m6A erasers, FTO and ALKBH5, was identified from the proteomic profiling of PCa cells (Figure 1). We utilized the AR-positive androgen-dependent LNCaP and DUCaP cell lines to represent early-stage hormone-sensitive (castrate-sensitive) PCa and the AR-negative androgen-independent PC3 and DU145 cell lines and the low-AR-expressing C4-2B cell line to represent advanced-stage metastatic CRPC [28]. Proteomic changes in between these cell lines were assessed using the SWATH-MS label-free quantification method (Appendix A).

The hierarchical clustering analysis demonstrated distinct clustering of the hormone/androgen-dependent and androgen-independent PCa cell lines based on the expression of m6A regulators (Figure 1A).

We further investigated the transcriptomic changes in the PCa cell lines, identifying eight differentially expressed m6A regulators. Genes such as *METTL3*, *IGF2BP2*, *YTHDF1*, *HNRNPC*, and *HNRNPA2B1* exhibited upregulation, while *WTAP*, *NUDT21*, and *FTO* showed downregulation in the androgen-independent cell lines compared to the androgen-dependent cell lines (Figure 1B). These results indicate the differential m6A regulator profiling of the PCa cell lines based on the PCa progression from the androgen-dependent to androgen-independent stage.

### 3.2. Effects of Androgen and Anti-Androgen Treatment on the Expression of m6A Methylation Regulators in PCa

Building upon the prior literature establishing a positive correlation between AR expression and m6A regulator expression, our investigation aimed to elucidate the connection between AR signaling activation and m6A regulator expression. The LNCaP PCa cell line was cultivated under androgen-deprived conditions and subjected to treatments involving androgen/DHT, as well as two widely used AR antagonists/anti-androgens in ATT: BIC andENZ. The SWATH-MS and RNAseq analyses revealed treatment-induced proteomic and transcriptomic changes in the m6A regulators (Appendix A).

The hierarchical analysis unveiled distinct clustering patterns in the m6A expression profiles induced by the androgen/DHT and anti-androgen treatments (Figure 2). Consistent with the existing literature, the heatmap analysis highlighted a limited correlation between the proteomic and transcriptomic expression data in the LNCaP cell model [22,29]. Notably, a pronounced increase in both transcriptomic and proteomic expression was observed for the m6A writers (METTL3, VIRMA, RBM15, RBM15B) and m6A readers (IGF2BP3, YTHDF1/2, YTHDC2, HNRNPA2B1, RBMX, PRRC2A, SRSF10, TRMT112), along with the m6A eraser ALKBH5, in response to the DHT treatment (Figure 2).

Interestingly, elevated proteomic expression was noted for the m6A writers (METTL14, WTAP, ZC3H13), m6A readers (YTHDF3, YTHDC1, XRN1, FMR1), and the m6A eraser FTO in response to the BIC treatment, both with and without DHT supplementation (Figure 2A).

### 3.3. Differential Expression of m6A Methylation Regulators in Primary PCa Tissue Using RNAseq

Subsequently, we conducted an analysis of the m6A regulator expression in primary PCa utilizing our RNAseq data derived from clinical specimens (Appendix A) [24]. The hierarchical clustering analysis, applied to the eight sets of primary PCa tumor and adjacent normal tissue RNAseq data, revealed distinct clustering of the primary PCa tumors and normal tissues, with the exception of the normal sample N_P2, based on the m6A regulator expression profiles (Figure 3A). The heatmap observations indicated the upregulated expression of all the m6A writers except *ZC3H13* in the primary tumor samples compared to the normal tissues (Figure 3A). In addition, the m6A readers, including *YTHDF1/2/3*, *HNRNPC*, *HNRNPA2B1*, *RBMX*, *PRRC2A*, and *CPSF6*, also demonstrated a higher expression in the tumor samples compared to the normal tissues (Figure 3A). On the other hand, the m6A eraser FTO showed decreased expression in most of the tumor samples compared to the normal tissues (Figure 3A). The principal component analysis further indicated similar m6A regulator expression profiles in the adjacent normal prostate tissues compared to the diverse and variable m6A expression profiles observed in the primary PCa tumor tissues (Figure 3B).

Validation through qRT-PCR analysis of 35 primary PCa tumor and adjacent normal tissues further confirmed the differential m6A regulator expression patterns in response to an advancing tumor grade. Specifically, the m6A writers *METTL3*, *WTAP*, and *RBM15B* and the m6A readers *YTHDF1* and *HNRNAPA2B1* showed significant upregulation in the high PCa tumor grades compared to the low tumor grades and adjacent normal tissues (Figure 3C). In contrast, the m6A writer *ZC3H13*, m6A reader *NUDT21*, and m6A eraser *FTO* were found to be significantly downregulated in the high-grade tumors compared to the low-grade tumors and adjacent normal tissues (Figure 3C).

### 3.4. Differential Expression of m6A Methylation Regulators in Response to Disease Progression

Using microarray transcriptomic data obtained from the publicly available PCa clinical dataset [27], we analyzed the m6A regulator expression in normal prostate tissues (*n* = 28), localized PCa tumor tissues (*n* = 59), and mCRPC tissues (*n* = 35) (Appendix A). The heatmap analyses demonstrated the differential m6A regulator expression patterns between the mCRPC tissues compared to the localized tumor and normal tissues (Figure 4A). The expression data highlighted that three m6A writers were differentially expressed in the mCRPC tissues, where the *METTL3* and *WTAP* genes showed significant downregulation and the *RBM15* gene showed significant upregulation in the mCRPC tissues compared to both the localized tumor and normal tissues. The analysis further indicated that three m6A readers are differentially expressed in the mCRPC tissues, where the *YTHDF2* gene showed significant upregulation, while the *NUDT21* and *SRSF10* genes showed significant downregulation in mCRPC compared to both the localized tumor and normal tissues (Figure 4B). Furthermore, significant downregulation of the m6A eraser, *FTO*, was observed in mCRPC compared to both the localized tumor tissues and normal tissues.

### 3.5. Survival Analysis in CRPC Patients Correlating with m6A Methylation Regulator Expression Levels

Univariate Kaplan–Meier survival analysis was conducted on the publicly available PCa microarray dataset [27] to assess the prognostic significance of m6A regulator gene expression in the survival of mCRPC patients following initial hormone therapy (Appendix A). The analysis revealed six m6A regulators with a significant association between gene expression and the survival outcomes of the mCRPC patients who underwent hormone therapy. The patients exhibiting a high expression of m6A readers, including *HNRNPC* (HR: 0.24, 95% CI 0.07–0.81, *p* = 0.013), *FMR1* (HR = 0.28, 95% CI 0.08–0.92, *p* = 0.024), *NUDT21* (HR = 0.26, 95% CI 0.07–0.97, *p* = 0.032), *RBMX* (HR = 0.2, 95% CI 0.06–0.72, *p* = 0.006), and *XRN1* (HR = 0.2, 95% CI 0.04–0.95, *p* = 0.026), had a better prognosis after hormone refractory therapy (Figure 5). Conversely, a higher expression of the m6A reader *IGF2BP3* (HR = 9.64, 95% CI: 2.06–45.2, *p* = 5 × 10^−4^) showed a statistically significant association with poorer survival in the mCRPC patients (Figure 5).

Notably, the RNA m6A writers (*METTL3*, *VIRMA*, *RBM15*, *METTL14*, *WTAP*, *RBM15B*, *ZC3H13*), m6A readers (*IGF2BP2*, *YTHDF1/2/3*, *YTHDC1/2*, *HNRNPA2B1*, *PRRC2A*, *CPSF6*, *NXF1*, *SRSF10*, *TRMT112*), and the m6A erasers FTO and ALKBH5 exhibited no significant association with the survival of the mCRPC patients undergoing hormone refractory therapy (Appendix A).

### 3.6. PCa-Risk-Associated Single-Nucleotide Polymorphisms (SNPs) and the Expression of m6A Methylation Regulators

To analyze the m6A regulators’ association with PCa-risk-associated SNPs, TCGA PCa patient tumor and normal tissue gene expression data were extracted and reanalyzed to identify eQTL. A total of 70 SNPs was found to be associated with PCa (2 of which were identified using genome-wide association studies) and showed correlation with the expression levels of 11 m6A regulator genes: *WTAP*, *IGF2BP2/3*, *YTHDC1/2*, *HNRNPC*, *HNRNPA2B1*, *PRRC2A*, *CPSF6*, *XRN1*, and *FTO* (Appendix A).

The SNPs with the strongest *p*-value of association for a specific gene were selected, and the corresponding genotypes for these SNPs and gene expression were correlated using ANOVA. Noteworthy findings included significant changes in the expression levels of *WTAP*, *FTO*, and *HNRNPA2B1* based on their homozygous and heterozygous genotypes in PCa. The most robust association was observed between *WTAP* and rs315987 (*p* = 5.56 × 10^−5^, FDR = 0.059), with the alternate allele (GG) linked to higher *WTAP* levels in PCa. Concurrently, SNP rs12537401 demonstrated a significant association with the *HNRNPA2B1* expression levels (*p* = 1.41 × 10^−4^, FDR = 0.043), wherein the recessive genotype (AA) exhibited elevated expression levels compared to the common genotype (CC). Additionally, a significant association between *FTO* and rs57038875 (*p* = 2.57 × 10^−4^, FDR = 0.036) was identified, with the recessive genotype (GG) correlated with lower expression levels of *FTO* (Figure 6).

Consistent with these findings, a substantial increase in expression was observed for *WTAP* and *HNRNPA2B1* in the primary TCGA PCa tumors, while *FTO* expression was downregulated in the TCGA primary PCa tumors compared to the adjacent normal tissues (Appendix A).

## 4. Discussion

Our investigation provides an in-depth exploration of the intricate landscape of RNA m6A methylation, a prominent epitranscriptomic modification within the context of PCa. Recent recognition of m6A’s pivotal roles in mRNA regulation, stability, and translation has sparked interest in its potential involvement in tumorigenesis, tumor development, and clinical applications [6,7,15,19].

Many previous studies investigated the changes at the transcriptomic level, and limited focus was given to the proteomic expression of the m6A regulators, which closely correlates with their cellular function [30,31,32,33]. The current study unveils complex relationships between the transcriptional activity and protein expression of m6A methylation regulators, revealing little concordance between the transcriptomics and proteomics data. This underscores nuanced interplay, which requires a deeper understanding, particularly concerning m6A methylation regulators and PCa.

Unlike previous studies that have primarily focused on the role of m6A regulators in primary PCa [30,31,32,33,34], our research pioneers an investigation into the characteristic changes in m6A regulator expression during the progression to CRPC. Leveraging SWATH mass spectrometry, the current study meticulously profiled the proteomic expression of 27 m6A regulators in both hormone/androgen-dependent and -independent PCa cell lines. Examining the impact of ATT on the m6A regulator signature revealed the differential behavior of m6A regulators in response to different anti-androgen treatments in PCa. M6A regulators that show dysregulation in response to androgen/DHT treatment could be directly or indirectly regulated by androgen/AR receptor signaling. These genes may be involved in PCa progression via elevated AR signaling activation during castration resistance. It is possible that treatments may induce distinct m6A methylation programming in cancer cells, which leads to certain alterations in tumor-specific oncogene expression or their function, thereby promoting tumor progression. According to our findings, the genes that were found to be upregulated under the stimulation of anti-androgen treatments may represent m6A regulators (i) that develop adaptive resistant mechanisms against therapy response or (ii) that augment therapy-induced tumor-killing mechanisms. Thus, deeper functional characterization of m6A regulators in PCa will assist in developing targeted therapies that block oncogenic regulators or developing gene therapies to induce tumor-killing regulators.

Transcriptomic data analysis of the PCa clinical specimens indicated the dysregulation of m6A regulators during PCa tumorigenesis and the emergence of castration resistance. However, the majority of the m6A regulators demonstrated showed no correlation between the PCa cell lines and the PCa clinical sample analysis. This observation points to the need for a larger panel of cancer cell lines and large clinical datasets to capture the true genetic heterogeneity of PCa. Nevertheless, the upregulation of *METTL3*, *YTHDF1*, and *HNRNPA2B1* and the downregulation of *NUDT21* and *FTO* were found to be associated with an advancing disease status, as evidenced by both the cell line and PCa clinical data.

Our observations of elevated *METTL3* levels in the advanced-stage PCa cell lines and high-grade PCa tumor samples are in line with previous evidence that showed the upregulation of *METTL3* in advanced-stage PCa [35,36]. These studies further demonstrate the oncogenic role of *METTL3* in promoting PCa growth of m6A-dependent stability regulation and the function of several oncogenic factors, such as MYC (C-myc) [16], LEF1 [37], and integrin β1 (ITGB1) [35,36]. However, the *METTL3* levels were found to be decreased in the mCRPC tissues compared to the localized tumors, as seen previously [38]. The studies demonstrate the association between a lower expression of *METTL3* and anti-androgen therapy resistance triggered by the upregulation of the nuclear receptor NR5A2/LRH-1 [38]. Similar to *METTL3* expression, *WTAP* also showed upregulation in the PCa tumors compared to the normal tissues and downregulation during PCa progression to mCRPC. The *WTAP* expression was previously reported to be higher in primary prostate adenocarcinoma patient samples compared with non-malignant prostate tissue [39]. However, further studies are warranted to delineate the functional role of *WTAP* during therapy resistance and PCa progression to mCRPC.

We observed significant upregulation of *RBM15B*, *YTHDF1*, and *HNRNAPA2B1* in the primary PCa tumors compared to the normal tissues and the upregulation of *RBM15* and *YTHDF2* in the mCRPC tissues compared to the localized tumors. Previous TCGA data analysis has highlighted significant upregulation of *RBM15B* in patient samples with high Gleason scores [30]. Our observations of elevated *YTHDF1* expression in high-grade tumors is in line with previous reports that demonstrate the association between high *YTHDF1* expression and a poor prognosis in PCa patients [40]. *HNRNAPA2B1* has been previously identified to be significantly associated with a high Gleason score and metastatic PCa disease [41].

Our analysis further indicated the downregulation of *NUDT21* and *FTO* in the primary PCa tumors compared to the normal tissues and downregulation in the mCRPC tissues compared to the localized tumors. The *NUDT21* gene, which encodes CFIm25 (a regulator of alternative polyadenylation), can play both tumor suppressor and oncogenic roles depending on the cancer type, yet limited studies have described its m6A-dependent role in cancers [42]. Our findings show *NUDT21* may be associated with a poor prognosis in PCa patients. In line with our findings, a previous study also showed decreased expression of *FTO* in advanced-tumor-stage and high-Gleason-score clinical samples [20]. In addition, *ZC3H13* expression was previously reported to have significantly lower expression in TCGA PCa clinical samples than normal samples, reiterating our findings [30]. Interestingly, another study has reported that *ZC3H13* can promote m6A-mediated expression of the tumor suppressor A1BG-AS1, highlighting the anti-tumor role of *ZC3H13* in PCa [43].

Several studies have attempted to characterize the diagnostic and prognostic potential of m6A regulators in PCa [30,32,44]. Two studies have provided intriguing evidence that an m6A score strategy developed based on m6A regulator expression can be utilized as a predictive biomarker for PCa metastasis [31,34]. Crucially, our study also identified six m6A readers, *HNRNPC*, *FMR1*, *NUDT21*, *RBMX*, *XRN1*, and *IGFBP3*, with suggestive evidence of prognostic value in CRPC through univariate Cox regression analyses. *HNRNPC* has been previously identified as a potential marker for a poor prognosis in many cancers, including PCa [45,46]. FMR1 has been previously identified as a promoter of PCa progression via the m6A-mediated circRBM33-FMR1 complex and has also been found to be correlated with a poor prognosis in the disease-free survival of PCa patients [47]. *RBMX* is known to control AR transcript metabolism by monitoring the turnover, synthesis, and splicing of AR variants and has been found to be correlated with a high Gleason score in PCa patients [48]. In contrast to the above findings, our results indicated that higher expression of *HNRNPC*, *FMR1*, and *RBMX* may have a protective effect aiding the survival of mCRPC patients. These findings indicate that m6A regulators may have distinct biological roles stimulated upon therapy response, either promoting or suppressing PCa progression to CRPC. *XRN1* has been previously reported to show different prognostic value in clinical PCa depending on the expression of its regulator, *miR-204*, highlighting its dual role in PCa [49]. The oncogenic role of *IGF2BP3* in tumorigenesis and tumor progression has been reported by previous studies, confirming our observations of *IGF2BP3*’s correlation with a poor prognosis in mCRPC patients [50].

Investigating the m6A regulator gene expression in TCGA PCa patients unveiled intriguing associations with PCa-risk-associated SNPs. Significant correlations have been observed between specific SNPs and elevated expression levels of *WTAP*, HNRNPA2B1, and *FTO.* Previous studies have reported the oncogenic role of *WTAP* and *HNRNPA2B1* and the tumor suppressor role of FTO in PCa. *HNRNPA2B1* can promote PCa progression via the HNRNPA2B1/*miR-93-5p*/FRMD6 signaling axis in an m6A-dependent manner [41]. *WTAP* can regulate the m6A-dependent translation of the oncogenic EIF3C protein, which drives PCa metastasis via the MAPK signaling pathway [51]. On the other hand, *FTO* can attenuate PCa proliferation and metastasis by reducing the degradation of *CLIC4* mRNA in an m6A-dependent manner [52]. Therefore, it is of great interest to study the underlying mechanism of the relationship between gene polymorphism and m6A regulator expression, whose dysregulation is linked with PCa progression. While our evolving comprehension of m6A regulators has prompted novel therapeutic strategies in clinical cancer therapy, our study unravels significant discordance among various m6A methylation regulators in PCa, emphasizing intricate and diverse biological effects that necessitate comprehensive exploration. Nonetheless, the empirical results reported herein should be considered in the light of some limitations. For example, a large-scale clinical validation cohort is still required to further clarify our observed association of m6A regulators with PCa progression and their prognostic performance, in combination with the current PCa biomarkers (e.g., PSA, PCA3). Although our eQTL analysis sheds some light, the molecular mechanisms of the divergent expression of m6A regulators in PCa and its progression remain elusive. Although the m6A readers and erasers have been functionally characterized, the m6A-dependent mechanisms of most m6A readers have yet to be investigated in many cancers, including PCa. On the other hand, a previous study demonstrated the correlation between m6A methylation and immune cell infiltration in PCa [33]. Another study has shown that *HNRNPC* may activate the immune-suppressive tumor microenvironment by promoting PCa progression [53]. Hence, it is noteworthy to study the therapeutic potential of targeting upstream m6A regulators to activate the immune microenvironment in PCa.

## 5. Conclusions

In conclusion, our study marks a significant stride in unraveling the complexities of m6A regulation in PCa. The findings unveil aberrations in m6A regulator expression in PCa progression and their association with PCa risk loci. Recognizing the heterogeneity within the PCa phenotypes and varying responses to treatment, the potential of prognostic stratification based on m6A regulator expression emerges as a promising avenue for enhancing PCa diagnosis and therapeutic interventions. Our study sets the stage for future investigations, paving the way for precision diagnostics and therapeutics for aggressive treatment-resistant PCa.

## Figures and Tables

**Figure 1 cancers-16-01303-f001:**
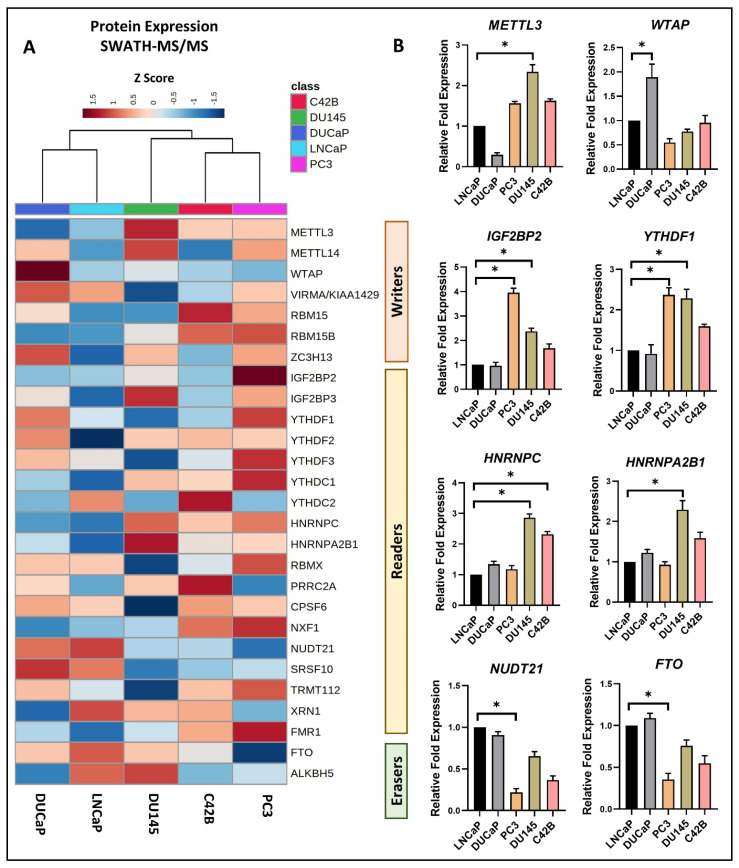
Differential expression of m6A methylation regulators in PCa cell lines examined using (**A**) SWATH-MS analysis and (**B**) qPCR analysis for proteomic and transcriptomic perspectives, respectively. (**A**) Normalized peak areas/intensities (Z Score) were utilized to generate a heatmap visualization of protein expression. Hierarchical clustering using the Ward algorithm, based on Euclidean distance, determined the column orders. (**B**) A total of eight m6A regulators exhibited significant differential expression in PCa cell lines. The relative fold expression of mRNA in each cell line was normalized to its expression in the LNCaP cell line using the ΔΔCT method (*n* = 3 technical replicates, mean ± SD, paired *t*-test * *p* < 0.05).

**Figure 2 cancers-16-01303-f002:**
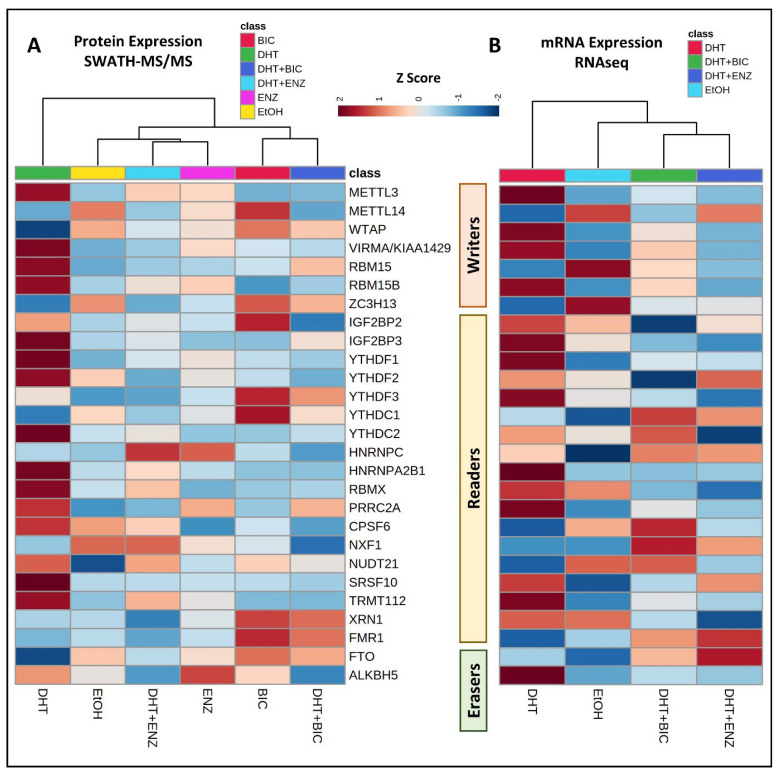
Hierarchical clustering was performed on the LNCaP cell line treated with androgen and anti-androgen, revealing distinct patterns in the differential expression of m6A methylation regulators. The analysis incorporated both proteomic and transcriptomic data obtained using (**A**) SWATH-MS analysis for protein expression and (**B**) RNAseq analysis for gene expression in response to androgen (DHT) and anti-androgen treatments, respectively. Heatmap visualization was facilitated by utilizing normalized peak areas/intensities and normalized Reads Per Kilobase Million (RPKM) for protein and mRNA expression, respectively. The hierarchical clustering employed the Ward algorithm, with column orders determined by applying Euclidean distance.

**Figure 3 cancers-16-01303-f003:**
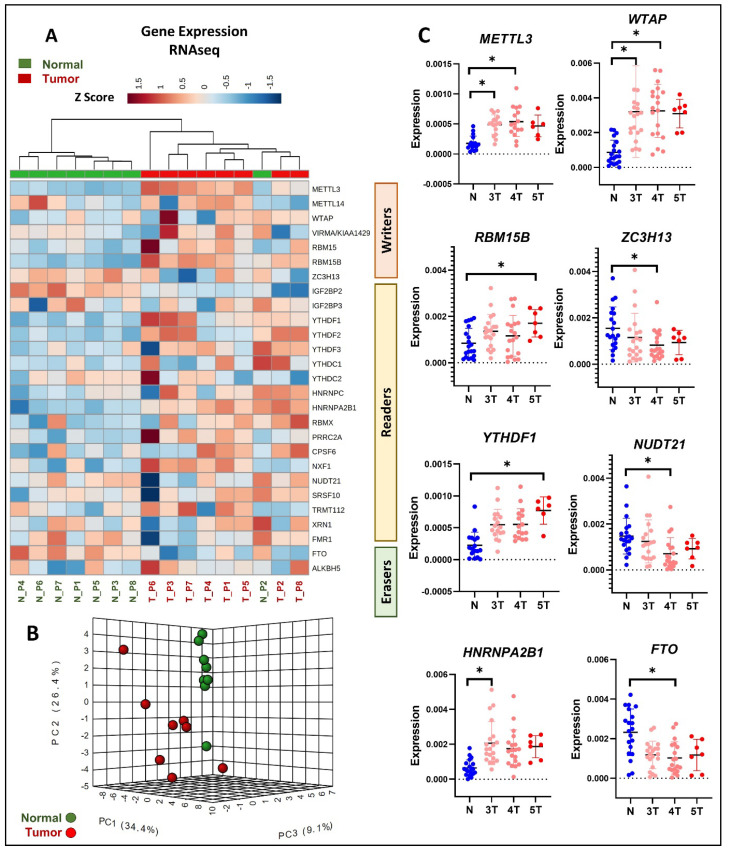
The expression of m6A regulator genes in primary PCa clinical tissues was assessed through both RNAseq and qRT-PCR analyses. (**A**) For RNAseq analysis, normalized m6A gene expression data were obtained from eight primary PCa clinical tissue samples and their adjacent normal tissues. Hierarchical clustering was performed using the Ward algorithm on the normalized RNAseq data, and column orders were arranged based on Euclidean distance. (**B**) Principal component analysis was conducted on the first three principal components identified from the normalized RNAseq data. (**C**) For qRT-PCR analysis, a total of 35 primary PCa clinical tissues with varying tumor grades, along with their adjacent normal tissue samples, were utilized. Tumor samples are denoted by red dots, showcasing an increasing intensity from low-grade to high-grade tumors, while blue dots represent adjacent normal tissues. Gene expression levels were determined using the ΔCT method (*n* = 3 technical replicates, mean ± SD, paired *t*-test * *p* < 0.05).

**Figure 4 cancers-16-01303-f004:**
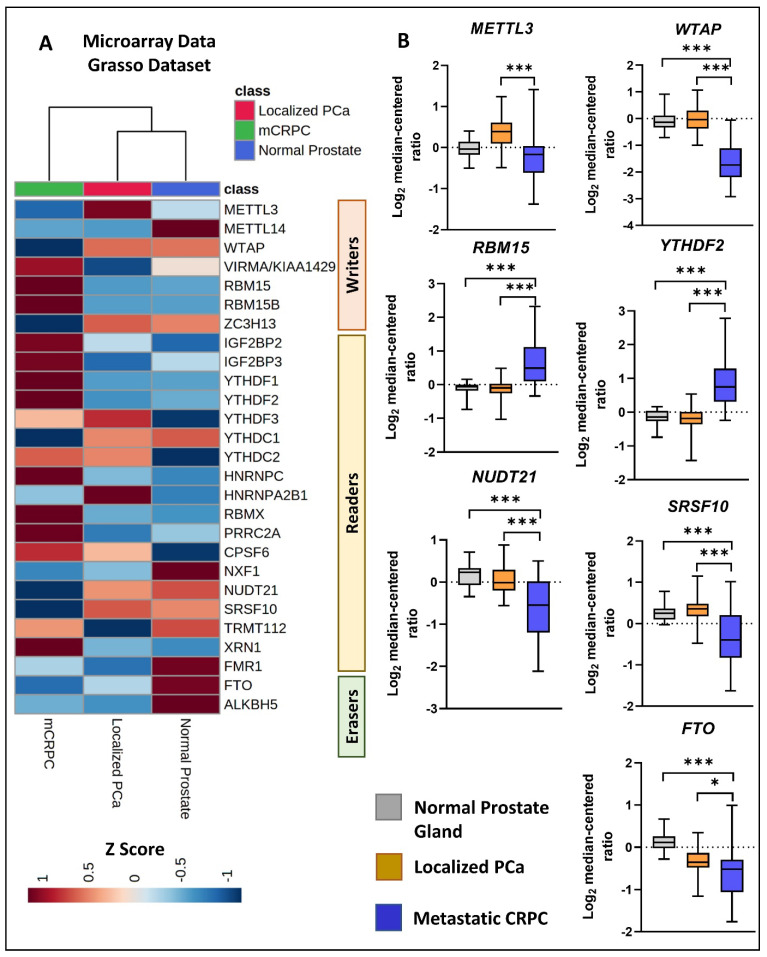
Differential expression of m6A methylation regulators in localized PCa and CRPC clinical tissues. (**A**) We conducted a comparison of the transcriptomic expression of m6A methylation regulators among non-PCa (*n* = 28), localized PCa (*n* = 59), and mCRPC (*n* = 35) clinical tissue specimens using tissue microarray data obtained from the previously published study [27]. The selected probes for the expression analysis are detailed in Appendix A. (**B**) The figure illustrates the expression of the top significantly differentially expressed microarray probes for each gene, as listed in Appendix A. Gene expression levels were measured as log_2_ median-centered ratios (mean ± SD, paired *t*-test, * *p* < 0.05, *** *p* < 0.0001).

**Figure 5 cancers-16-01303-f005:**
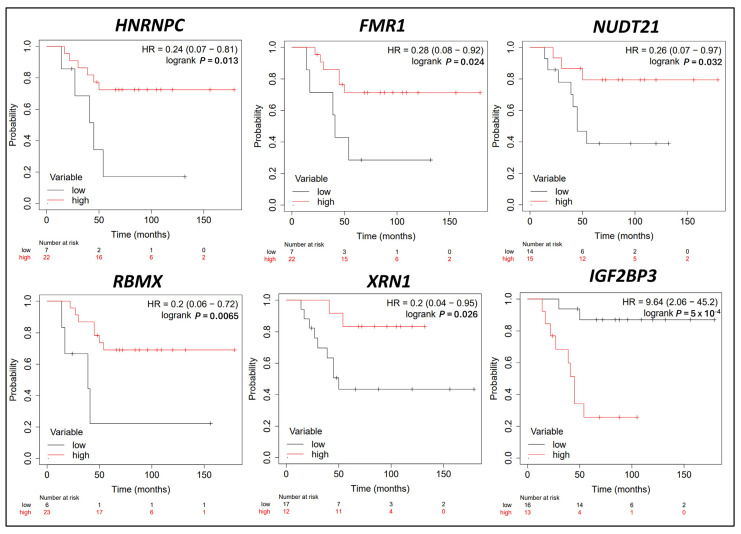
Univariate Kaplan–Meier survival analysis was employed to investigate the impact of m6A methylation regulators on the overall survival of CRPC patients following the initial hormone refractory therapy. A total of six m6A regulators demonstrated a significant correlation between their expression levels and the overall survival of CRPC patients post initial hormone refractory therapy. The survival times of CRPC patients (in months) and their survival status at 5 years can be found in Appendix A. Hazard Ratio (HR) denotes the risk associated with low or high expression of the m6A regulator. The *p*-values presented originate from log-rank tests comparing the two Kaplan–Meier curves, and significance was considered at *p* < 0.05.

**Figure 6 cancers-16-01303-f006:**
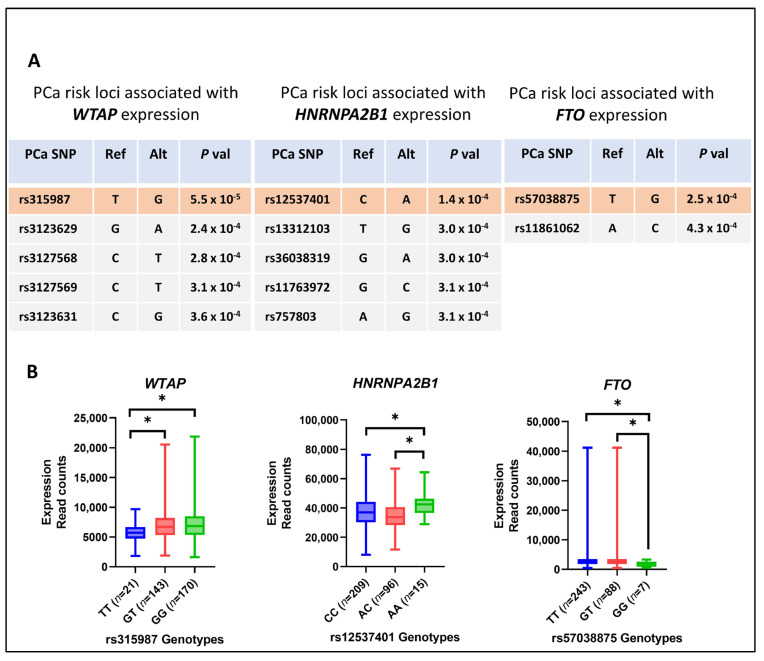
Association between PCa risk loci and differential expression of m6A methylation regulators in PCa. (**A**) Utilizing Cis-eQTL analysis, we identified three m6A regulators (*WTAP*, *HNRNPA2B1*, *FTO*) that exhibited a significant association with PCa risk loci. The top SNPs were shortlisted based on their significance (*p*-value) in association with the three m6A regulator genes. (**B**) illustrates the differential expression of the three m6A regulator genes in the presence of risk and wild-type genotypes of the most significantly associated PCa SNP, * *p* < 0.05.

## Data Availability

All data relevant to the publication has been presented in the Appendix A.

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
