# Peer review of "RNA m6a Methylation Regulator Expression in Castration-Resistant Prostate Cancer Progression and Its Genetic Associations"

_cancers, 2024, doi:10.3390/cancers16071303_

Round 1

Reviewer 1 Report

Comments and Suggestions for Authors

In the manuscript by Liyanage et al. the authors provide a comprehensive analysis of m6A regulator expression in PCa progression, shedding light on their connection with PCa risk loci.  Using SWATH mass spectrometry and RNAseq analysis, we conducted a thorough examination of the expression of 27 m6A regulators in hormone/androgen dependent and independent PCa cell lines. The findings showed clear differences in clustering between tumor and adjacent normal prostate tissues.  The data presented in the manuscript is interesting. The experiments are very well designed, and the manuscript was well written.  This will add an extra layer of information highlighting significant cross talk between RNA m6A methylation and prostate cancer progression.

I have a few suggestions for polishing the manuscript.

Major Comments:

1.     The authors present their findings using high throughput transcriptomics and proteomics approach.  The authors should rewrite the results section and should focus on the findings as well as what is the reason behind these findings.  At most places there are areas where there are a lot of data, and it is easy to lose track as the authors did not explain their results in an elaborate manner.  They should rewrite the section.

2.     The authors should re-validate their proteomics data using Western blot as they re-validated their RNA-seq data using RT-qPCR. This will additionally strengthen their data.

3.     In Page 2, Line 90-91, the authors say ‘where androgen-dependent cell lines showed overlapping clustering for the 1st and 2nd principal components’- is there any Figure they missed.

4.     In Figure 2B, the authors should include BIC and ENZ treatment data as well.

Minor Comments:

1.     The authors should add a graphical abstract to explain their results for the broader audience.

Author Response

On behalf of all authors, I would like to thank for Reviewer 1 comments and suggestions to better refine the manuscript.

In the manuscript by Liyanage et al. the authors provide a comprehensive analysis of m6A regulator expression in PCa progression, shedding light on their connection with PCa risk loci.  Using SWATH mass spectrometry and RNAseq analysis, we conducted a thorough examination of the expression of 27 m6A regulators in hormone/androgen dependent and independent PCa cell lines. The findings showed clear differences in clustering between tumor and adjacent normal prostate tissues.  The data presented in the manuscript is interesting. The experiments are very well designed, and the manuscript was well written.  This will add an extra layer of information highlighting significant cross talk between RNA m6A methylation and prostate cancer progression.

I have a few suggestions for polishing the manuscript.

Major Comments

  1. The authors present their findings using high throughput transcriptomics and proteomics approach.  The authors should rewrite the results section and should focus on the findings as well as what is the reason behind these findings.  At most places there are areas where there are a lot of data, and it is easy to lose track as the authors did not explain their results in an elaborate manner.  They should rewrite the section.

Authors appreciate reviewers’ suggestion on refining the results section. Results section has been modified in most section to better explain the findings. We have added more explanations in to the discussion section describing the reason behind the current results/findings/observations.

  1. The authors should re-validate their proteomics data using Western blot as they re-validated their RNA-seq data using RT-qPCR. This will additionally strengthen their data.

We acknowledge reviewers concerns on validating MS data using western blot. However, due to the unavailability of antibodies and time limitation of performing western blot for 27 different markers, we did not perform western blot analysis to validate the current MS data. This has been acknowledged in the revised manuscript.

  1. In Page 2, Line 90-91, the authors say ‘where androgen-dependent cell lines showed overlapping clustering for the 1st and 2nd principal components’- is there any Figure they missed.

Results description about principal component analysis has been removed. PCA plot has been removed from the Figure 1 as it replicates the findings in hierarchical clustering in heatmap analysis.

  1. In Figure 2B, the authors should include BIC and ENZ treatment data as well.

RNAseq analysis data was obtained from our previously published study (Lai, et al., Scientific Reports, 2017), which investigated KLK alternative transcripts expression in LNCaP cells in response to ATT treatments. As this original analysis did not perform BIC and ENZ alone treatments, we were restricted to DHT and DHT+ BIC/ENZ treatment groups to reanalyse the expression of analysis m6A regulator enzymes. Explanation of data reanalysis has been added to the methodology section. In the current study, mass-spectroscopy proteomic profiling of BIC/ENZ treatment with and without DHT treatment was performed to better refine the androgen or anti-androgen regulated protein expression in LNCaP cells.

Minor Comments

  1. The authors should add a graphical abstract to explain their results for the broader audience.

As per reviewers’ suggestion, a graphical abstract has been added to summarize the findings of the current study.

Reviewer 2 Report

Comments and Suggestions for Authors

The manuscript from Liyanage et al profiles the expression of RNA m6a writers, readers, and erasers in prostate cancer cell lines and tumor samples. This is an area of active research in prostate cancer. The findings presented are largely descriptive and, in many cases, confirmatory.  The manuscript does provide some interesting findings beyond what has been published by other investigators, but the amount of new data and how it relates to previously published work should be discussed in more detail in the Discussion section (see below). More specific comments are provided below.

1) Unlike other reports dealing with RNA m6a writers, readers, and erasers, which only analyzed RNA data, some of the results provided in this study include protein data along with RNA data. However, only relative data is presented (z scores). The actual data should be provided as supplementary data for figures 1 and 2. The authors should also provide the actual data for the RNA studies in figures 1, 2 and 3.

2) The authors should discuss how the data for prostate cancer cell lines compares with patient clinical data. For example, how similar are the data for androgen dependent cell lines and the patient data for primary prostate cancer? Similarly, how similar are the data for the androgen independent cell lines and the patient data for mCRPC?

3) The data in figure 2 demonstrate that the RNA m6a writers and readers are strongly upregulated when LNCaP cells grown in androgen depleted media and then treated with DHT. This novel finding was hardly explored. Had this been done, it would have substantially increased the impact of this paper. In addition, for at least a few genes, RNA and protein expression varied significantly. This should be verified by western blot analysis and discussed in either the Results section or the Discussion section. Also, how does the response to DHT relate to the expression of these gene in mCRPC.

4) The data in figure 3 is not overly novel. A quick look at published studies shows the data for five of the 8 genes described in part C has been established by other investigators. This may be true for the other three genes as well. How the data in this figure compares with published data needs to be discussed.

5) The prognostic value of two of the genes in figure 5 (FMR1 and IGF2BP3) has been reported by others. Although this is briefly noted in the Discussion section, further commentary is warranted. What about the other 4 genes?  Have they been identified as prognostic markers for patient survival by other investigators? In contrast, two genes (YTHDF2 and HNRNPA2B1) have been identified as prognostic markers by other investigators, but not in this study. Is there a reason for this? Please discuss.

6) The principal component analysis referred to on lines 89-91 is not provided.

7) In several instances, the writing could be improved. The most notable examples are lines 33-36; line 46 (have been, not: are); line 56 (These).

8) The authors’ use of the word pioneering is a bit over the top. Please remove the hype.

Comments on the Quality of English Language

In several instances, the writing could be improved. The most notable examples are lines 33-36; line 46 (have been, not: are); line 56 (These).

The authors’ use of the word pioneering is a bit over the top. Please remove the hype.

Author Response

On behalf of all the authors, I would like to express our sincere gratitude to Reviewer 2 for their valuable comments and suggestions, which have greatly contributed to refining the manuscript. The manuscript from Liyanage et al profiles the expression of RNA m6a writers, readers, and erasers in prostate cancer cell lines and tumor samples. This is an area of active research in prostate cancer. The findings presented are largely descriptive and, in many cases, confirmatory.  The manuscript does provide some interesting findings beyond what has been published by other investigators, but the amount of new data and how it relates to previously published work should be discussed in more detail in the Discussion section (see below). More specific comments are provided below.

  1. Unlike other reports dealing with RNA m6a writers, readers, and erasers, which only analyzed RNA data, some of the results provided in this study include protein data along with RNA data. However, only relative data is presented (z scores). The actual data should be provided as supplementary data for figures 1 and 2. The authors should also provide the actual data for the RNA studies in figures 1, 2 and 3.

According to Reviewer 2 suggestions, mass-spectroscopy and RNAseq expression data of m6A-regulator expression has been added to the supplementary table file.

Table S1. SWATH-MS label free quantification of m6A-regulator expression in prostate cancer cell lines; Table S2. SWATH-MS label free quantification of m6A-regulator expression in androgen and anti-androgen treated LNCaP cell line; Table S3. RNA sequencing transcriptomic analysis of m6A-regulator expression in androgen and anti-androgen treated LNCaP cell line; Table S4. RNA sequencing analysis of m6A-regulator expression in prostate cancer tumor and adjacent normal tissues.

2) The authors should discuss how the data for prostate cancer cell lines compares with patient clinical data. For example, how similar are the data for androgen dependent cell lines and the patient data for primary prostate cancer? Similarly, how similar are the data for the androgen independent cell lines and the patient data for mCRPC?

Below explanation has been added to discussion section to explaining the results of PCa cell line PCa clinical data in comparison with the findings of previous literature.

(Line 431-475)

Transcriptomic data analysis of tumor and adjacent normal prostate tissues in PCa clinical specimens indicated dysregulation based on the m6A regulators during PCa tumorigenesis and the emergence of castration resistance. However, the majority of the m6A regulators showed no correlation between PCa cell line and PCa clinical sample analysis. This observation underscores the need for a larger panel of cancer cell lines and large clinical datasets to capture the true genetic heterogeneity of PCa.

Nevertheless, upregulation of METTL3, YTHDF1, HNRNPA2B1, and downregulation of NUDT21 and FTO were found to be associated with advancing disease status, as evidenced by both cell line and PCa clinical data. Our observations of elevated METTL3 levels in advanced stage PCa cell lines and high-grade PCa tumor samples align with previous evidence showing upregulation of METTL3 in advanced stage PCa [34,35]. These studies further demonstrate its oncogenic role in promoting PCa growth by regulating the m6A-dependent stability and function of several oncogenic factors, such as MYC (c-myc) [16], LEF1 [36], Integrin β1 (ITGB1) [34,35]. However, METTL3 levels were found to be decreased in mCRPC tissues compared to localized tumors, similar to previous reports associating decreased METTL3 expression with mCRPC [37]. These studies highlight the association between lower METTL3 expression and anti-androgen therapy resistance triggered by the upregulation of the nuclear receptor NR5A2/LRH-1 [37].

Similar to METTL3 expression, WTAP also showed upregulation in PCa tumors compared to normal tissues and downregulation during PCa progression to mCRPC. WTAP expression was previously reported for its higher expression in primary prostate adenocarcinoma patient samples compared with non-malignant prostate tissue [38]. However, further studies are warranted to delineate the functional role of WTAP during therapy resistance and PCa progression to mCRPC.

According to the analysis, there was significant upregulation of RBM15B, YTHDF1, HNRNAPA2B1 in primary PCa tumors compared to normal tissues and upregulation of RBM15, YTHDF2 in mCRPC tissues compared to localized tumors. Previous TCGA data analysis has highlighted significant upregulation of RBM15B in patient samples with a high Gleason score [29]. Our observations of elevated YTHDF1 expression in high-grade tumors align with previous reports demonstrating the association between high YTHDF1 expression and poor prognosis for PCa patients [39]. HNRNAPA2B1 has been previously identified as significantly associated with a high Gleason score and metastatic PCa disease [40].

Our analysis further indicated the downregulation of NUDT21, FTO in primary PCa tumors compared to normal tissues and downregulation in mCRPC tissues compared to localized tumors. The NUDT21 gene that encodes CFIm25 (a regulator of alternative polyadenylation) can play both tumor-suppressor and oncogenic roles depending on the cancer type, yet limited studies have described its m6A-dependent role in cancers [41]. Our findings suggest that NUDT21 may be associated with the poor prognosis of PCa patients, yet further studies are required to delineate the functional role of NUDT21 in PCa.

In line with our findings, a previous study also showed decreased expression of FTO in high tumor stage and high Gleason score clinical samples [20]. Additionally, ZC3H13 expression was previously reported to be significantly lower in TCGA PCa clinical samples than in normal samples, reinforcing our findings [29]. Interestingly, another study has reported that ZC3H13 can promote m6A-mediated expression of the tumor suppressor A1BG-AS1, highlighting the anti-tumor role of ZC3H13 in PCa [42].)

3) The data in figure 2 demonstrate that the RNA m6a writers and readers are strongly upregulated when LNCaP cells grown in androgen depleted media and then treated with DHT. This novel finding was hardly explored. Had this been done, it would have substantially increased the impact of this paper. In addition, for at least a few genes, RNA and protein expression varied significantly. This should be verified by western blot analysis and discussed in either the Results section or the Discussion section. Also, how does the response to DHT relate to the expression of these gene in mCRPC.

We acknowledge reviewers concerns on validating MS data using western blot. However, due to the time limitation and budget constraints of ordering antibodies and time limitation of performing western blot for 27 different markers, we did not perform western blot analysis to validate the current MS data.

The below section has been added to discussion section to explain importance of genes in the context of mCRPC that dysregulate in response to DHT and anti-androgen treatments.

(Line 418-429)

M6A regulators that show dysregulation in response to androgen/DHT treatment could be directly or indirectly regulated by the androgen/AR-receptor signaling. These genes may involve in PCa progression via elevated AR-signaling activation during castra-tion-resistance. It is possible that treatments may induce distinct m6A-methylation programming in cancer cells that leads to certain alterations in tumor-specific oncogene expression or their function thereby promoting tumor progression. According to our findings, genes that found to be upregulated under stimulation of anti-androgen treatments may represent m6A-regulators (i) that develop adaptive resistant mechanisms against therapy response or (ii) that augment the therapy induced tumor-killing mechanisms. Thus, deeper functional characterization of m6A-regulators in PCa will assist to develop targeted therapies that block oncogenic regulators or to develop gene-therapies to induce tumor-killing regulators.

As per our previous observations (Liyanage et al., 2020, Cancers) and in agreement with published research from previous researchers (Ferrari et al., 2017, Cell Communication and Signalling, Iglesias-Gato et al., 2016, European urology) we observed a limited correlation between PCa transcriptomic and proteomic datasets. Varied post-transcriptional modifications (PTMs) associated with translation regulation and kinetic differences between protein synthesis and turnover may explain the poor correlation between mRNA and protein abundances in complex biological samples.

4) The data in figure 3 is not overly novel. A quick look at published studies shows the data for five of the 8 genes described in part C has been established by other investigators. This may be true for the other three genes as well. How the data in this figure compares with published data needs to be discussed.

Please refer to the response to question number 2. A discussion on previous reports of m6A-regulator expression in clinical samples comparing to those identified in the current study, have been incorporated.

5) The prognostic value of two of the genes in figure 5 (FMR1 and IGF2BP3) has been reported by others. Although this is briefly noted in the Discussion section, further commentary is warranted. What about the other 4 genes?  Have they been identified as prognostic markers for patient survival by other investigators? In contrast, two genes (YTHDF2 and HNRNPA2B1) have been identified as prognostic markers by other investigators, but not in this study. Is there a reason for this? Please discuss.

Prognostic value of FMR1 and IGF2BP3 has been further added in discussion.

Line 481-496

HNRNPC has been previously identified as a potential marker for the poor-prognosis of many cancers including PCa [45,46]. FMR1 has been previously identified as a promoter of PCa progression via m6A-mediated circRBM33-FMR1 complex and also found correlated with the poor-prognosis of the disease-free survival of PCa patients that activates mitochondrial metabolism in PCa cells [47]. RBMX is known to control AR transcript metabolism by monitoring turnover, synthesis, and splicing of AR-variants and found correlated with the high-Gleason score of PCa patients [48]. In contrast to above findings our results indicated that HNRNPC, FMR1 and RBMX higher expression may have a protective effect towards the survival of mCRPC patients. These findings indicate that m6A-regulators may have distinct biological roles stimulated upon therapy-response promoting or suppressing PCa progression to CRPC. XRN1 has been previously reported to show different prognostic value in clinical PCa depending on its regulator: miR-204 expression, highlighting its dual role in PCa [49]. Oncogenic role of IGF2BP3 in tumorigenesis and tumor progression has been reported by previous studies, confirming our observations of IGF2BP3 correlation with the poor-prognosis of mCRPC patients [50].,

6) The principal component analysis referred to on lines 89-91 is not provided.

The PCA plot has been excluded from Figure 1 as it duplicates the observations obtained from the hierarchical clustering in heatmap analysis. Reference to these results in the revised manuscript has been appropriately removed.7) In several instances, the writing could be improved. The most notable examples are lines 33-36; line 46 (have been, not: are); line 56 (These).

We appreciate reviewers’ suggestions on the quality of language use. We have made changes to writing to improve the language quality.

(Line 47-53)

Due to the PCa dependence on oncogenic androgen receptor (AR) signaling, androgen deprivation therapy (ADT) is considered as the primary systemic therapy for patients with localized, recurrent, or advanced-stage disease. ADT is, often supplemented with androgen-targeted therapy (ATT) which uses anti-androgen treatments to suppress the activation of androgen receptor (AR) signaling [3].

(Line 62-66)

Presently, over 170 post-synthetic RNA nucleoside modifications form the human 'epitranscriptome' [10]. These RNA modifications play a crucial role in regulating post-transcriptional stability, localization, translocation, splicing, and translation, contributing to various biological processes and diseases [11].

(Line 70-75)

Like DNA methylation, m6A methylation is reversibly regulated by enzymes: m6A methyltransferases (METTL3/14, WTAP, RBM15/15B, VIRMA, and ZC3H13, termed 'writers'), and m6A demethylases (FTO, ALKBH5, and ALKBH3, termed 'erasers'). M6A methylation execute its primary function via interaction with RNA binding proteins (YTHDC1/2, YTHDF1/2/3, IGF2BP1/2/3, HNRNPs, termed 'readers').

8) The authors’ use of the word pioneering is a bit over the top. Please remove the hype.

As per reviewers’ suggestions we have removed the word pioneering.

(Line 517-518)

Nonetheless, the empirical results reported herein should be considered in the light of some limitations.

Comments on the Quality of English Language

In several instances, the writing could be improved. The most notable examples are lines 33-36; line 46 (have been, not: are); line 56 (These).

The authors’ use of the word pioneering is a bit over the top. Please remove the hype.

We appreciate reviewers’ suggestions on the quality of language. We have made changes to writing to improve the language quality.

Please refer to the responses given in Question 7 and 8.

Round 2

Reviewer 1 Report

Comments and Suggestions for Authors

In the manuscript by Liyanage et al. the authors provide a comprehensive analysis of m6A regulator expression in PCa progression, shedding light on their connection with PCa risk loci.  Using SWATH mass spectrometry and RNAseq analysis, we conducted a thorough examination of the expression of 27 m6A regulators in hormone/androgen dependent and independent PCa cell lines. The findings showed clear differences in clustering between tumor and adjacent normal prostate tissues.  The data presented in the manuscript is interesting. The experiments are very well designed, and the manuscript was well written.  This will add an extra layer of information highlighting significant cross talk between RNA m6A  methylation and prostate cancer progression.

The authors have addressed all the previous comments. Thus, the manuscript can be accepted in its present form.

Reviewer 2 Report

Comments and Suggestions for Authors

The manuscript is improved significantly.

Comments on the Quality of English Language

Minor editing is needed for lines 418-429 and lines 481-496.